# RIDE: Rewarding Impact-Driven Exploration for Procedurally-Generated Environments

**Roberta Raileanu**[*]
Facebook AI Research
New York University
raileanu@cs.nyu.edu

**Tim Rocktäschel**
Facebook AI Research
University College London
rockt@fb.com

## Abstract

Exploration in sparse reward environments remains one of the key challenges of model-free reinforcement learning. Instead of solely relying on extrinsic rewards provided by the environment, many state-of-the-art methods use intrinsic rewards to encourage exploration. However, we show that existing methods fall short in procedurally-generated environments where an agent is unlikely to visit a state more than once. We propose a novel type of intrinsic reward which encourages the agent to take actions that lead to significant changes in its learned state representation. We evaluate our method on multiple challenging procedurally-generated tasks in MiniGrid, as well as on tasks with high-dimensional observations used in prior work. Our experiments demonstrate that this approach is more sample efficient than existing exploration methods, particularly for procedurally-generated MiniGrid environments. Furthermore, we analyze the learned behavior as well as the intrinsic reward received by our agent. In contrast to previous approaches, our intrinsic reward does not diminish during the course of training and it rewards the agent substantially more for interacting with objects that it can control.

## 1 Introduction

Deep reinforcement learning (RL) is one of the most popular frameworks for developing agents that can solve a wide range of complex tasks (Mnih et al., 2016; Silver et al., 2016; 2017). RL agents learn to act in new environments through trial and error, in an attempt to maximize their cumulative reward. However, many environments of interest, particularly those closer to real-world problems, do not provide a steady stream of rewards for agents to learn from. In such settings, agents require many episodes to come across any reward, often rendering standard RL methods inapplicable.

Inspired by human learning, the use of intrinsic motivation has been proposed to encourage agents to learn about their environments even when extrinsic feedback is rarely provided (Schmidhuber, 1991b; 2010; Oudeyer et al., 2007; Oudeyer & Kaplan, 2009). This type of exploration bonus emboldens the agent to visit new states (Bellemare et al., 2016; Burda et al., 2019b; Ecoffet et al., 2019) or to improve its knowledge and forward prediction of the world dynamics (Pathak et al., 2017; Burda et al., 2019a), and can be highly effective for learning in hard exploration games such as Montezuma's Revenge (Mnih et al., 2016). However, most hard exploration environments used in previous work have either a limited state space or an easy way to measure similarity between states (Ecoffet et al., 2019) and generally use the same "*singleton*" environment for training and evaluation (Mnih et al., 2016; Burda et al., 2019a). Deep RL agents trained in this way are prone to overfitting to a specific environment and often struggle to generalize to even slightly different settings (Rajeswaran et al., 2017; Zhang et al., 2018a;b). As a first step towards addressing this problem, a number of procedurally-generated environments have been recently released, for example DeepMind Lab (Beattie et al., 2016), Sokoban (Racanière et al., 2017), Malmö (Johnson et al., 2016), CraftAssist (Jernite et al., 2019), Sonic (Nichol et al., 2018), CoinRun (Cobbe et al., 2019), Obstacle Tower (Juliani et al., 2019), or Capture the Flag (Jaderberg et al., 2019).

In this paper, we investigate exploration in procedurally-generated sparse-reward environments. Throughout the paper, we will refer to the general problem that needs to be solved as the *task*

---

[*]Work done during an internship at Facebook AI Research.

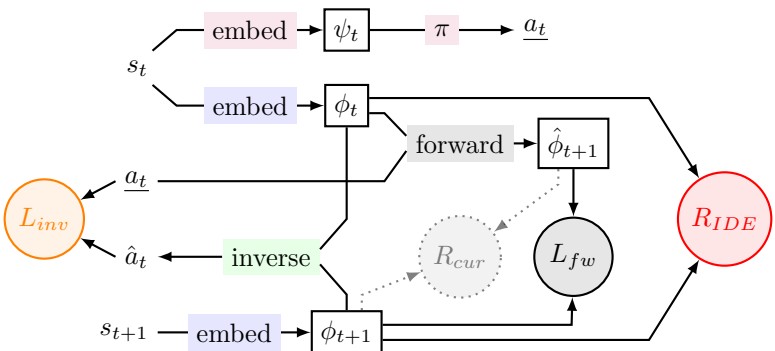

Figure 1: RIDE rewards the agent for actions that have an impact on the state representation ($R_{IDE}$), which is learned using both a forward ($L_{fw}$) and an inverse dynamics ($L_{inv}$) model.

(*e.g.* find a goal inside a maze) and to the particular instantiation of this task as the *environment* (*e.g.* maze layout, colors, textures, locations of the objects, environment dynamics etc.). The environment can be *singleton* or *procedurally-generated*. Singleton environments are those in which the agent has to solve the same task in the same environment in every episode, *i.e.*., the environment does not change between episodes. A popular example of a hard exploration environment that falls into that category is Montezuma's Revenge. In procedurally-generated environments, the agent needs to solve the same task, but in every episode the environment is constructed differently (*e.g.* resulting in a different maze layout), making it unlikely for an agent to ever visit the same state twice. Thus, agents in such environments have to learn policies that generalize well across a very large state space. We demonstrate that current exploration methods fall short in such environments as they (i) make strong assumptions about the environment (deterministic or resettable to previous states) (Ecoffet et al., 2019; Aytar et al., 2018), (ii) make strong assumptions about the state space (small number of different states or easy to determine if two states are similar) (Ecoffet et al., 2019; Burda et al., 2019b; Bellemare et al., 2016; Ostrovski et al., 2017; Machado et al., 2018a), or (iii) provide intrinsic rewards that can diminish quickly during training (Pathak et al., 2017; Burda et al., 2019a).

To overcome these limitations, we propose **Rewarding Impact-Driven Exploration** (RIDE), a novel intrinsic reward for exploration in RL that encourages the agent to take actions which result in impactful changes to its representation of the environment state (see Figure 1 for an illustration). We compare against state-of-the-art intrinsic reward methods on singleton environments with high-dimensional observations (*i.e.* visual inputs), as well as on hard-exploration tasks in procedurally-generated grid-world environments. Our experiments show that RIDE outperforms state-of-the-art exploration methods, particularly in procedurally-generated environments. Furthermore, we present a qualitative analysis demonstrating that RIDE, in contrast to prior work, does not suffer from diminishing intrinsic rewards during training and encourages agents substantially more to interact with objects that they can control (relative to other state-action pairs).

## 2 RELATED WORK

The problem of exploration in reinforcement learning has been extensively studied. Exploration methods encourage RL agents to visit novel states in various ways, for example by rewarding surprise (Schmidhuber, 1991b;a; 2010; 2006; Achiam & Sastry, 2017), information gain (Little & Sommer, 2013; Still & Precup, 2012; Houthooft et al., 2016), curiosity (Pathak et al., 2017; Burda et al., 2019b), empowerment (Klyubin et al., 2005; Rezende & Mohamed, 2015; Gregor et al., 2017), diversity (Eysenbach et al., 2019), feature control (Jaderberg et al., 2017; Dilokthanakul et al., 2019), or decision states (Goyal et al., 2019; Modhe et al., 2019). Another class of exploration methods apply the Thompson sampling heurisitc (Osband et al., 2016; Ostrovski et al., 2017; O'Donoghue et al., 2018; Tang et al., 2017). Osband et al. (2016) use a family of randomized Q-functions trained on bootstrapped data to select actions, while Fortunato et al. (2018) add noise in parameter space to encourage exploration. Here, we focus on intrinsic motivation methods, which are widely-used and have proven effective for various hard-exploration tasks (Mnih et al., 2016; Pathak et al., 2017; Bellemare et al., 2016; Burda et al., 2019b).

Intrinsic motivation can be useful in guiding the exploration of RL agents, particularly in environments where the extrinsic feedback is sparse or missing altogether (Oudeyer et al., 2007; 2008; Oudeyer & Kaplan, 2009; Schmidhuber, 1991b; 2010). The most popular and effective kinds of intrinsic motivation can be split into two broad classes: count-based methods that encourage the agent to visit novel states and curiosity-based methods that encourage the agent to learn about the environment dynamics.

**Count-Based Exploration.** Strehl & Littman (2008) proposed the use of state visitation counts as an exploration bonus in tabular settings. More recently, such methods were extended to high-dimensional state spaces (Bellemare et al., 2016; Ostrovski et al., 2017; Martin et al., 2017; Tang et al., 2017; Machado et al., 2018a). Bellemare et al. (2016) use a Context-Tree Switching (CTS) density model to derive a state pseudo-count, while Ostrovski et al. (2017) use PixelCNN as a state density estimator. Burda et al. (2019b) employ the prediction error of a random network as exploration bonus with the aim of rewarding novel states more than previously seen ones. However, one can expect count-based exploration methods to be less effective in procedurally-generated environments with sparse reward. In these settings, the agent is likely to characterize two states as being different even when they only differ by features that are irrelevant for the task (*e.g.* the texture of the walls). If the agent considers most states to be "novel", the feedback signal will not be distinctive or varied enough to guide the agent.

**Curiosity-Driven Exploration.** Curiosity-based bonuses encourage the agent to explore the environment to learn about its dynamics. Curiosity can be formulated as the error or uncertainty in predicting the consequences of the agent's actions (Stadie et al., 2015; Pathak et al., 2017; Burda et al., 2019b). For example, Pathak et al. (2017) learn a latent representation of the state and design an intrinsic reward based on the error of predicting the next state in the learned latent space. While we use a similar mechanism for learning state embeddings, our exploration bonus is very different and builds upon the difference between the latent representations of two consecutive states. As we will see in the following sections, one problem with their approach is that the intrinsic reward can vanish during training, leaving the agent with no incentive to further explore the environment and reducing its feedback to extrinsic reward only.

**Generalization in Deep RL.** Most of the existing exploration methods that have achieved impressive results on difficult tasks (Ecoffet et al., 2019; Pathak et al., 2017; Burda et al., 2019b; Bellemare et al., 2016; Choi et al., 2019; Aytar et al., 2018), have been trained and tested on the same environment and thus do not generalize to new instances. Several recent papers (Rajeswaran et al., 2017; Zhang et al., 2018a;b; Machado et al., 2018b; Foley et al., 2018) demonstrate that deep RL is susceptible to severe overfitting. As a result, a number of benchmarks have been recently released for testing generalization in RL (Beattie et al., 2016; Cobbe et al., 2019; Packer et al., 2018; Justesen et al., 2018; Leike et al., 2017; Nichol et al., 2018; Juliani et al., 2019). Here, we make another step towards developing exploration methods that can generalize to unseen scenarios by evaluating them on procedurally-generated environments. We opted for MiniGrid (Chevalier-Boisvert et al., 2018) because it is fast to run, provides a standard set of tasks with varied difficulty levels, focuses on single-agent, and does not use visual inputs, thereby allowing us to better isolate the exploration problem.

More closely related to our work are the papers of Marino et al. (2019) and Zhang et al. (2019). Marino et al. (2019) use a reward that encourages changing the values of the non-proprioceptive features for training low-level policies on locomotion tasks. Their work assumes that the agent has access to a decomposition of the observation state into internal and external parts, an assumption which may not hold in many cases and may not be trivial to obtain even if it exists. Zhang et al. (2019) use the difference between the successor features of consecutive states as intrinsic reward. In this framework, a state is characterized through the features of all its successor states. While both of these papers use fixed (*i.e.* not learned) state representations to define the intrinsic reward, we use forward and inverse dynamics models to learn a state representation constrained to only capture elements in the environment that can be influenced by the agent. Lesort et al. (2018) emphasize the benefits of using a learned state representation for control as opposed to a fixed one (which may not contain information relevant for acting in the environment). In the case of Zhang et al. (2019), constructing a temporally extended state representation for aiding exploration is not trivial. Such a feature space may add extra noise to the intrinsic reward due to the uncertainty of future states. This is particularly problematic when the environment is highly stochastic or the agent often encounters novel states (as it is the case in procedurally-generated environments).

## 3  BACKGROUND: CURIOSITY-DRIVEN EXPLORATION

We use the standard formalism of a single agent Markov Decision Process (MDP) defined by a set of states $\mathcal{S}$, a set of actions $\mathcal{A}$, and a transition function $\mathcal{T} : \mathcal{S} \times \mathcal{A} \rightarrow \mathcal{P}(\mathcal{S})$ providing the probability distribution of the next state given a current state and action. The agent chooses actions by sampling from a stochastic policy $\pi : \mathcal{S} \rightarrow \mathcal{P}(\mathcal{A})$, and receives reward $r : \mathcal{S} \times \mathcal{A} \rightarrow \mathbb{R}$ at every time step. The agent's goal is to learn a policy which maximizes its discounted expected return $R_t = \mathbb{E}\left[\sum_{k=0}^{T} \gamma^k r_{t+k+1}\right]$ where $r_t$ is the sum of the intrinsic and extrinsic reward received by the agent at time $t$, $\gamma \in [0, 1]$ is the discount factor, and the expectation is taken with respect to both the policy and the environment. Here, we consider the case of episodic RL in which the agent maximizes the reward received within a finite time horizon.

In this paper we consider that, along with the extrinsic reward $r_t^e$, the agent also receives some intrinsic reward $r_t^i$, which can be computed for any $(s_t, a_t, s_{t+1})$ tuple. Consequently, the agent tries to maximize the weighted sum of the intrinsic and extrinsic reward: $r_t = r_t^e + \omega_{ir} r_t^i$ where $\omega_{ir}$ is a hyperparameter to weight the importance of both rewards.

We built upon the work of Pathak et al. (2017) who note that some parts of the observation may have no influence on the agent's state. Thus, Pathak et al. propose learning a state representation that disregards those parts of the observation and instead only models (i) the elements that the agent can control, as well as (ii) those that can affect the agent, even if the agent cannot have an effect on them. Concretely, Pathak et al. learn a state representations $\phi(s) = f_{emb}(s; \theta_{emb})$ of a state $s$ using an inverse and a forward dynamics model (see Figure 1). The forward dynamics model is a neural network parametrized by $\theta_{fw}$ that takes as inputs $\phi(s_t)$ and $a_t$, predicts the next state representation: $\hat{\phi}(s_{t+1}) = f_{fw}(\phi_t, a_t; \theta_{fw})$, and it is trained to minimize $L_{fw}(\theta_{fw}, \theta_{emb}) = \|\hat{\phi}(s_{t+1}) - \phi(s_{t+1})\|_2^2$. The inverse dynamics model is also a neural network parameterized by $\theta_{inv}$ that takes as inputs $\phi(s_t)$ and $\phi(s_{t+1})$, predicts the agent's action: $\hat{a}_t = f_{inv}(\phi_t, \phi_{t+1}; \theta_{inv})$, and it is trained to minimize $L_{inv}(\theta_{inv}, \theta_{emb}) = CrossEntropy(\hat{a}_t, a_t)$ when the action space is discrete. Pathak et al.'s curiosity-based intrinsic reward is proportional to the squared Euclidean distance between the actual embedding of the next state $\phi(s_{t+1})$ and the one predicted by the forward model $\hat{\phi}(s_{t+1})$.

## 4  IMPACT-DRIVEN EXPLORATION

Our main contribution is a novel intrinsic reward based on the change in the state representation produced by the agent's action. The proposed method encourages the agent to try out actions that have a significant impact on the environment. We demonstrate that this approach can promote effective exploration strategies when the feedback from the environment is sparse.

We train a forward and an inverse dynamics model to learn a latent state representation $\phi(s)$ as proposed by Pathak et al. (2017). However, instead of using the Euclidean distance between the predicted next state representation and the actual next state representation as intrinsic reward ($R_{cur}$ in Figure 1), we define impact-driven reward as the Euclidean distance between consecutive state representations ($R_{IDE}$ in Figure 1). Compared to curiosity-driven exploration, impact-driven exploration rewards the agent for very different state-actions, leading to distinct agent behaviors which we analyze in Section 6.1.1.

Stanton & Clune (2018) categorize exploration into: *across-training* and *intra-life* and argue they are complementary. Popular methods such as count-based exploration (Bellemare et al., 2016) encourage agents to visit novel states in relation to all prior training episodes (*i.e.* across-training novelty), but they do not consider whether an agent visits novel states within some episode (*i.e.* intra-life novelty). As we will see, RIDE combines both types of exploration.

Formally, RIDE is computed as the $L_2$-norm $\|\phi(s_{t+1}) - \phi(s_t)\|_2$ of the difference in the learned state representation between consecutive states. However, to ensure that the agent does not go back and forth between a sequence of states (with a large difference in their embeddings) in order to gain intrinsic reward, we discount RIDE by episodic state visitation counts. Concretely, we divide the impact-driven reward by $\sqrt{N_{ep}(s_{t+1})}$, where $N_{ep}(s_{t+1})$ is the number of times that state has been visited during the current episode, which is initialized to 1 in the beginning of the episode. In high-

dimensional regimes, one can use episodic pseudo-counts instead (Bellemare et al., 2016; Ostrovski et al., 2017). Thus, the overall intrinsic reward provided by RIDE is calculated as:

$$R_{IDE}(s_t, a_t) \equiv r_t^i(s_t, a_t) = \frac{\|\phi(s_{t+1}) - \phi(s_t)\|_2}{\sqrt{N_{ep}(s_{t+1})}}$$

where $\phi(s_{t+1})$ and $\phi(s_t)$ are the learned representations of consecutive states, resulting from the agent transitioning to state $s_{t+1}$ after taking action $a_t$ in state $s_t$. The state is projected into a latent space using a neural network with parameters $\theta_{emb}$.

The overall optimization problem that is solved for training the agent is

$$\min_{\theta_\pi, \theta_{inv}, \theta_{fw}, \theta_{emb}} [\omega_\pi L_{RL}(\theta_\pi) + \omega_{fw} L_{fw}(\theta_{fw}, \theta_{emb}) + \omega_{inv} L_{inv}(\theta_{inv}, \theta_{emb})]$$

where $\theta_\pi$ are the parameters of the policy and value network ($a_t \sim \pi(s_t; \theta_\pi)$), and $\omega_\pi$, $\omega_{inv}$ and $\omega_{fw}$ are scalars that weigh the relative importance of the reinforcement learning (RL) loss to that of the inverse and forward dynamics losses which are used for learning the intrinsic reward signal. Note that we never update the parameters of the inverse ($\theta_{inv}$), forward ($\theta_{fw}$), or embedding networks ($\theta_{emb}$) using the signal from the intrinsic or extrinsic reward (*i.e.* the RL loss); we only use these learned state embeddings for constructing the exploration bonus and never as part of the agent's policy (Figure 1 highlights that the policy learns its own internal representation of the state $\psi_t$, which is only used for control and never for computing the intrinsic reward). Otherwise, the agent can artificially maximize its intrinsic reward by constructing state representations with large distances among themselves, without grounding them in environment observations.

Note that there is no incentive for the learned state representations to encode features of the environment that cannot be influenced by the agent's actions. Thus, our agent will not receive rewards for reaching states that are inherently unpredictable, making exploration robust with respect to distractor objects or other inconsequential sources of variation in the environment. As we will later show, RIDE is robust to the well-known noisy-TV problem in which an agent, that is rewarded for errors in the prediction of its forward model (such as the one proposed in Pathak et al. (2017)), gets attracted to local sources of entropy in the environment. Furthermore, the difference of consecutive state representations is unlikely to go to zero during learning as they are representations of actual states visited by the agent and constrained by the forward and inverse model. This is in contrast to Pathak et al. (2017) and Burda et al. (2019b) where the intrinsic reward goes to zero as soon as the forward model becomes sufficiently accurate or the agent's policy only explores well known parts of the state space.

## 5 EXPERIMENTS

We evaluate RIDE on procedurally-generated environments from MiniGrid, as well as on two existing singleton environments with high-dimensional observations used in prior work, and compare it against both standard RL and three commonly used intrinsic reward methods for exploration. For all our experiments, we show the mean and standard deviation of the average return across 5 different seeds for each model. The average return is computed as the rolling mean over the past 100 episodes.

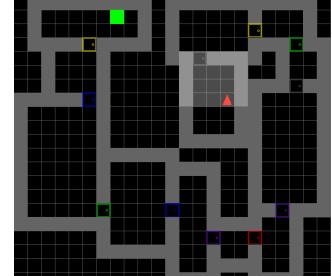

### 5.1 ENVIRONMENTS

The first set of environments are procedurally-generated gridworlds in MiniGrid (Chevalier-Boisvert et al., 2018). We consider three types of hard exploration tasks: *MultiRoomNXSY*, *KeyCorridorS3R3*, and *ObstructedMaze2Dlh*.

Figure 2: Rendering of a procedurally-generated environment from MiniGrid's MultiRoomN12S10 task.

In MiniGrid, the world is a partially observable grid of size $N \times N$. Each tile in the grid contains at most one of the following objects: wall, door, key, ball, box and goal. The agent can take one of seven actions: turn left or right, move forward, pick up or drop an object, toggle or done. More details about the MiniGrid environment and tasks can be found in A.3.

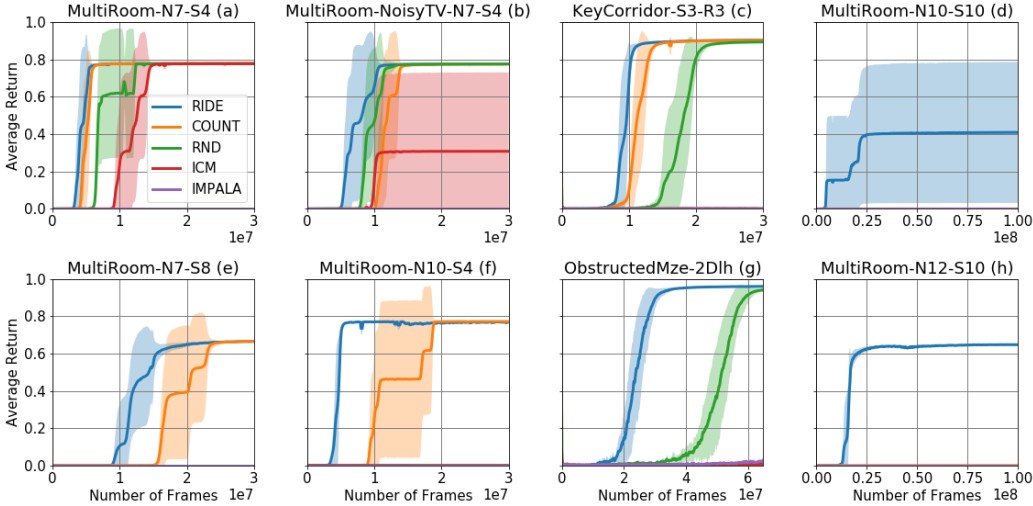

Figure 3: Performance of RIDE, Count, RND, ICM and IMPALA on a variety of hard exploration problems in MiniGrid. Note RIDE is the only one that can solve the hardest tasks.

For the sole purpose of comparing in a fair way to the curiosity-driven exploration work by Pathak et al. (2017), we ran a one-off experiment on their Mario (singleton) environment (Kauten, 2018). We train our model with and without extrinsic reward on the first level of the game.

The last (singleton) environment we evaluate on is *VizDoom* (Kempka et al., 2016). Details about the environment can be found in A.4.

## 5.2 BASELINES

For all our experiments, we use IMPALA (Espeholt et al., 2018) following the implementation of Küttler et al. (2019) as the base RL algorithm, and RMSProp (Tieleman & Hinton, 2012) for optimization. All models use the same basic RL algorithm and network architecture for the policy and value functions (see Appendix A.2 and Appendix A.1 for details regarding the hyperparameters and network architectures), differing only in how intrinsic rewards are defined. In our experiments we compare with the following baselines: **Count**: Count-Based Exploration by Bellemare et al. (2016) which uses state visitation counts to give higher rewards for new or rarely seen states. **RND**: Random Network Distillation Exploration by Burda et al. (2019b) which uses the prediction error of a random network as exploration bonus with the aim of rewarding novel states more than previously encountered ones. **ICM**: Intrinsic Curiosity Module by Pathak et al. (2017) (see Section 3). **IMPALA**: Standard RL approach by Espeholt et al. (2018) that uses only extrinsic reward and encourages random exploration by entropy regularization of the policy.

## 6 RESULTS AND DISCUSSION

We present the results of RIDE in comparison to popular exploration methods, as well as an analysis of the learned policies and properties of the intrinsic reward generated by different methods.

### 6.1 MINIGRID

Figure 3 summarizes our results on various hard MiniGrid tasks. Note that the standard RL approach IMPALA (purple) is not able to learn in any of the environments since the extrinsic reward is too sparse. Furthermore, our results reveal that RIDE is more sample efficient compared to all the other exploration methods across all MiniGrid tasks considered here. While other exploration bonuses seem effective on easier tasks and are able to learn optimal policies where IMPALA fails, the gap between our approach and the others is increasing with the difficulty of the task. Furthermore, RIDE manages to solve some very challenging tasks on which the other methods fail to get any reward even after training on over 100M frames (Figure 3).

**RND**            **ICM**            **RIDE**

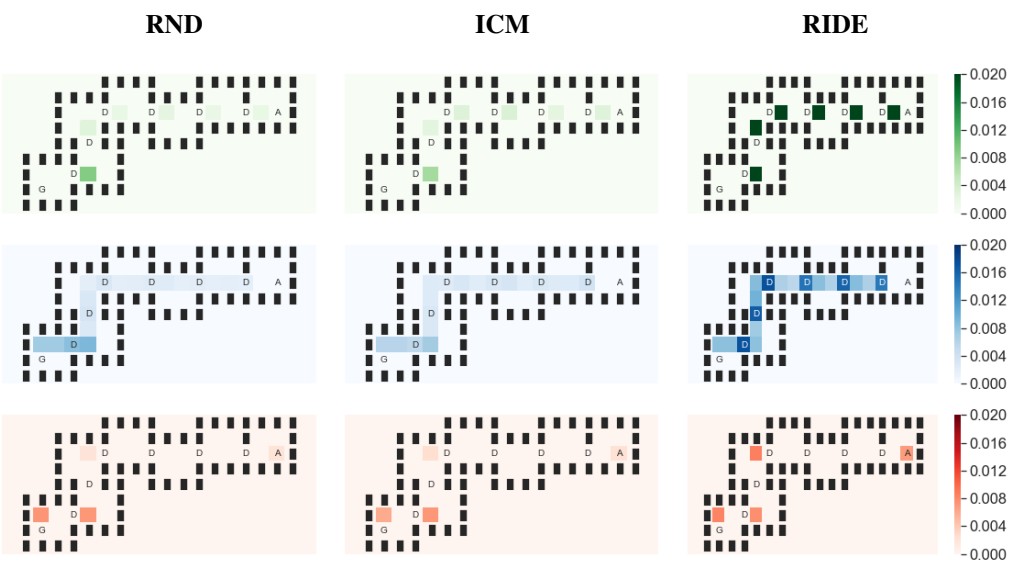

Figure 4: Intrinsic reward heatmaps for RND, ICM, and RIDE (from left to right) for opening doors (green), moving forward (blue), or turning left or right (red) on a random environment from the MultiRoomN7S4 task. A is the agent's starting position, G is the goal position and D are doors that have to be opened on the way.

| Model | Open Door | | Turn Left / Right | | Move Forward | |
|---|---|---|---|---|---|---|
| | Mean | Std | Mean | Std | Mean | Std |
| RIDE | 0.0490 | 0.0019 | 0.0071 | 0.0034 | 0.0181 | 0.0116 |
| RND | 0.0032 | 0.0018 | 0.0031 | 0.0028 | 0.0026 | 0.0017 |
| ICM | 0.0055 | 0.0003 | 0.0052 | 0.0003 | 0.0056 | 0.0003 |

Table 1: Mean intrinsic reward per action over 100 episodes on a random maze in MultiRoomN7S4.

In addition to existing MiniGrid tasks, we also tested the model's ability to deal with stochasticity in the environment by adding a "noisy TV" in the MiniGridN7S4 task, resulting in the new Mini-GirdN7S4NoisyTV task (left-center plot in the top row of Figure 3). The noisy TV is implemented as a ball that changes its color to a randomly picked one whenever the agent takes a particular action. As expected, the performance of ICM drops as the agent becomes attracted to the ball while obtaining intrinsic rewarded for not being able to predict the next color. The Count model also needs more time to train, likely caused by the increasing number of rare and novel states (due to the changing color of the ball).

We include results for ablations to our model in Appendix A.5, highlighting the importance of combining impact-driven exploration with episodic state visitation discounting.

### 6.1.1 ANALYSIS OF THE INTRINSIC REWARD

To better understand the effectiveness of different exploration methods, we investigate the intrinsic reward an agent receives for certain trajectories in the environment.

Figure 4 shows a heatmap of the intrinsic reward received by RND, ICM, and RIDE on a sampled environment after having been trained on procedurally-generated environments from the MultiRoomN7S4 task. While all three methods can solve this task, the intrinsic rewards received are different. Specifically, the RIDE agent is rewarded in a much more structured manner for opening doors, entering new rooms and turning at decision points. Table 1 provides quantitative numbers for this phenomenon. We record the intrinsic rewards received for each type of action, averaged over 100 episodes. We found that RIDE is putting more emphasis on actions interacting with the door than for moving forward or turning left or right, while the other methods reward actions more uniformly.

Figure 12 and Table 3 in A.6.2 show a similar pattern for the intrinsic rewards for agents trained on the MultiRoomN12S10 task, while Figure 13 and Table 4 in A.6.3 contain the equivalent analysis for agents trained on ObstructedMaze2Dlh. As emphasized there, RIDE is rewarding the agent more for interactions with objects as opposed to actions for moving around in the maze, a characteristic which is not as prevalent in the other models.

Figure 5 shows the mean intrinsic reward of all models while training on the MultiRoomN12S10 task. While the ICM, RND, and Count intrinsic reward converges to very low values quite early in the training process, the RIDE bonus keeps changing and has a higher value even after training on 100M frames. Hence, RIDE constantly encourages the agent to take actions that change the local environment. In contrast, Count, RND, and Curiosity may not consider certain states to be "novel" or "surprising" after longer periods of training as they have seen similar states in the past or learned to almost perfectly predict the next state in a subset of the environment states. Consequently, their intrinsic rewards diminish during training and the agent struggles to distinguish between actions that lead to novel or surprising states from those that do not, thereby getting trapped in some parts of the state space (see Figure 12).

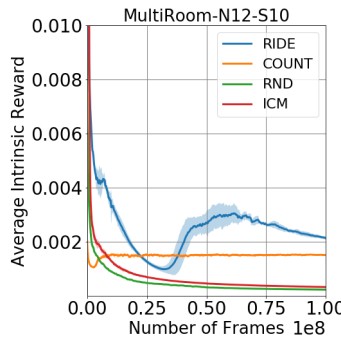

Figure 5: Mean intrinsic reward for models trained on Multi-RoomN12S10.

### 6.1.2 SINGLETON VERSUS PROCEDURALLY-GENERATED ENVIRONMENTS

It is important to understand and quantify how much harder it is to train existing deep RL exploration methods on tasks in procedurally-generated environments compared to a singleton environment.

To investigate this dependency, we trained the models on a singleton environment of the the *ObstructedMaze2Dlh* task so that at the beginning of every episode, the agent is spawned in exactly the same maze with all objects located in the same positions. In this setting, we see that Count, RND, and IM-PALA are also able to solve the task (see Figure 6 and compare with the center-right plot in the bottom row of Figure 3 for procedurally-generated environments of the same task). As expected, this emphasizes that training an agent in procedurally-generated environments creates significant challenges over training on a singleton environment for the same task. Moreover, it highlights the importance of training on a variety of environments to avoid overfitting to the idiosyncrasies of a particular environment.

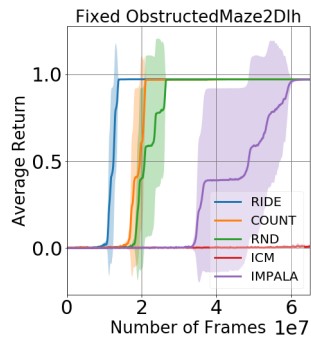

Figure 6: Training on a singleton instance of ObstructedMaze2Dlh.

### 6.1.3 NO EXTRINSIC REWARD

To analyze the way different methods explore environments without depending on the chance of running into extrinsic reward (which can dramatically change the agent's policy), we analyze agents that are trained without any extrinsic reward on both singleton and procedurally-generated environments.

The top row of Figure 7 shows state visitation heatmaps for all the models in a singleton environment on the MultiRoomN10S6 task, after training all of them for 50M frames with intrinsic reward only. The agents are allowed to take 200 steps in every episode. The figure indicates that all models have effective exploration strategies when trained on a singleton maze, the 10th, 9th and 6th rooms are reached by RIDE, Count/RND, and ICM, respectively. The Random policy fully explores the first room but does not get to the second room within the time limit.

When trained on procedurally-generated mazes, existing models are exploring much less efficiently as can be seen in the bottom row of Figure 7. Here, Count, RND, and ICM only make it to the 4th, 3rd and 2nd rooms respectively within an episode, while RIDE is able to explore all rooms. This

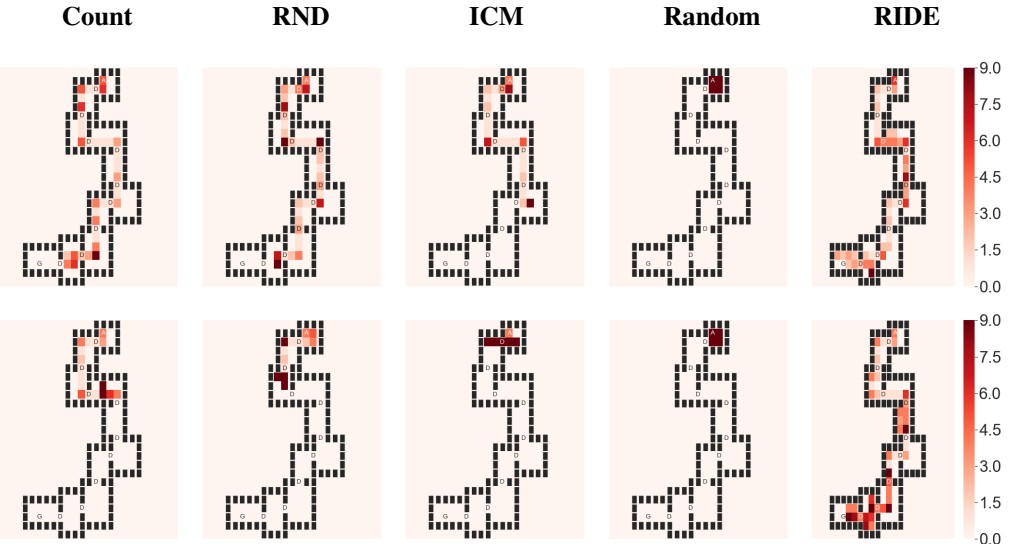

Figure 7: State visitation heatmaps for Count, RND, ICM, Random, and RIDE models (from left to right) trained for 50m frames without any extrinsic reward on a singleton maze (top row) and on procedurally-generated mazes (bottom row) in MultiRoomN10S6.

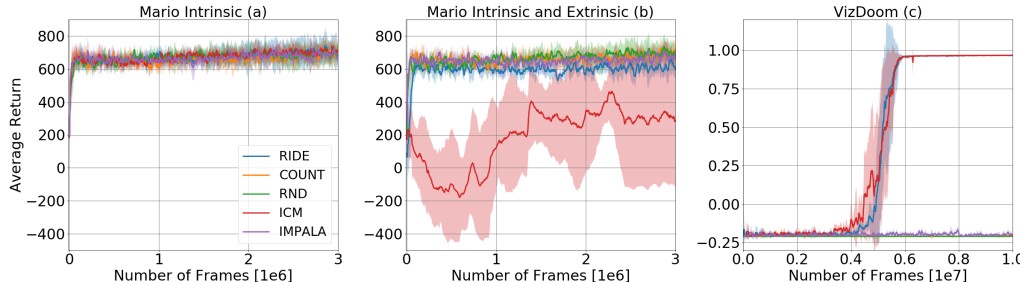

Figure 8: Performance on Mario with intrinsic reward only (a), with intrinsic and extrinsic reward (b), and VizDoom (c). Note that IMPALA is trained with extrinsic reward only in all cases.

further supports that RIDE learns a state representation that allows generalization across different mazes and is not as distracted by less important details that change from one procedurally-generated environment to another.

## 6.2 MARIO AND VIZDOOM

In order to compare to Pathak et al. (2017), we evaluate RIDE on the first level of the Mario environment. Our results (see Figure 8 a and b) suggest that this environment may not be as challenging as previously believed, given that all the methods evaluated here, including vanilla IMPALA, can learn similarly good policies after training on only 1m frames even without any intrinsic reward (left figure). Note that we are able to reproduce the results mentioned in the original ICM paper (Pathak et al., 2017). However, when training with both intrinsic and extrinsic reward (center figure), the curiosity-based exploration bonus (ICM) hurts learning, converging later and to a lower value than the other methods evaluated here.

For VizDoom (see Figure 8 c) we observe that RIDE performs as well as ICM, while all the other baselines fail to learn effective policies given the same amount of training. Note that our ICM implementation can reproduce the results in the original paper on this task, achieving a $100\%$ success rate after training on approximately 60m frames (Pathak et al., 2017).

## 7 CONCLUSION AND FUTURE WORK

In this work, we propose Rewarding Impact-Driven Exploration (RIDE), an intrinsic reward bonus that encourages agents to explore actions that substantially change the state of the environment, as measured in a learned latent space. RIDE has a number of desirable properties: it attracts agents to states where they can affect the environment, it provides a signal to agents even after training for a long time, and it is conceptually simple as well as compatible with other intrinsic or extrinsic rewards and any deep RL algorithm.

Our approach is particularly effective in procedurally-generated sparse-reward environments where it significantly outperforms IMPALA (Espeholt et al., 2018), as well as some of the most popular exploration methods such as Count (Bellemare et al., 2016), RND (Burda et al., 2019b), and ICM (Pathak et al., 2017). Furthermore, RIDE explores procedurally-generated environments more efficiently than other exploration methods.

However, there are still many ways to improve upon RIDE. For example, one can make use of symbolic information to measure or characterize the agent's impact, consider longer-term effects of the agent's actions, or promote diversity among the kinds of changes the agent makes to the environment. Another interesting avenue for future research is to develop algorithms that can distinguish between desirable and undesirable types of impact the agent can have in the environment, thus constraining the agent to act safely and avoid distractions (*i.e.* actions that lead to large changes in the environment but that are not useful for a given task). The different kinds of impact might correspond to distinctive skills or low-level policies that a hierarchical controller could use to learn more complex policies or better exploration strategies.

### ACKNOWLEDGMENTS

We would like to thank Heinrich Küttler, Edward Grefenstette, Nantas Nardelli, Jakob Foerster, Kyunghyun Cho, Arthur Szlam, Rob Fergus, Victor Zhong and Léon Bottou for insightful discussions and valuable feedback on this work.

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

## A  APPENDIX

### A.1  NETWORK ARCHITECTURES

All our models use the same network architecture for the policy and value networks. The input is passed through a sequence of three (for MiniGrid) or four (for the environments used by Pathak et al. (2017)) convolutional layers with 32 filters each, kernel size of 3x3, stride of 2 and padding of 1. An exponential linear unit (ELU; (Clevert et al. (2016))) is used after each convolution layer. The output of the last convolution layer is fed into a LSTM with 256 units. Two separate fully connected layers are used to predict the value function and the action from the LSTM feature representation.

For the singleton environments used in prior work, the agents are trained using visual inputs that are pre-processed similarly to Mnih et al. (2016). The RGB images are converted into gray-scale and resized to $42 \times 42$. The input given to both the policy and the state representation networks consists of the current frame concatenated with the previous three frames. In order to reduce overfitting, during training, we use action repeat of four. At inference time, we sample the policy without any action repeats.

### A.2  HYPERPARAMETERS

We ran grid searches over the learning rate $\in [0.0001, 0.0005, 0.001]$, batch size $\in [8, 32]$ and unroll length $\in [20, 40, 100, 200]$. The best values for all models can be found in Table 2. The learning rate is linearly annealed to 0 in all experiments.

| Parameter | Value |
|---|---|
| Learning Rate | 0.0001 |
| Batch Size | 32 |
| Unroll Length | 100 |
| Discount | 0.99 |
| RMSProp Momentum | 0.0 |
| RMSProp $\epsilon$ | 0.01 |
| Clip Gradient Norm $\epsilon$ | 40.0 |

Table 2: Hyperparameters common to all experiments.

We also ran grid searches over the intrinsic reward coefficient $\in$ [1.0, 0.5, 0.1, 0.05, 0.01, 0.005, 0.001] and the entropy coefficient $\in$ [0.01, 0.005, 0.001, 0.0005, 0.0001, 0.00005] for all the models on all environments. The best intrinsic reward coefficient was 0.1 for ICM and RND, and 0.005 for Count on all environments. The best entropy coefficient was 0.0001 for ICM, RND, and Count on all environments. For RIDE, we used an intrinsic reward coefficient of 0.1 and entropy coefficient of 0.0005 for MultiRoomN7S4, MultiRoomNoisyTVN7S4, MultiRoomN10S4, KeyCorridorS3R3 and 0.5, 0.001 for MultiRoomN7S8, MultiRoomN10S10, MultiRoomN12S10, and Obstructed-Maze2Dlh. The ablations use the same hyperparameters as RIDE. In all experiments presented here, we use the best values found for each model.

### A.3  MINIGRID ENVIRONMENT

In MiniGrid, the world is a partially observable grid of size NxN. Each tile in the grid contains exactly zero or one objects. The possible object types are wall, door, key, ball, box and goal.

Each object in MiniGrid has an associated discrete color, which can be one of red, green, blue, purple, yellow or grey. By default, walls are always grey and goal squares are always green. Rewards are sparse for all MiniGrid environments.

There are seven actions in MiniGrid: turn left, turn right, move forward, pick up an object, drop an object, toggle and done. The agent can use the turn left and turn right action to rotate and face one of 4 possible directions (north, south, east, west). The move forward action makes the agent move from its current tile onto the tile in the direction it is currently facing, provided there is nothing on

that tile, or that the tile contains an open door. The agent can open doors if they are right in front of it by using the toggle action.

Observations in MiniGrid are partial and egocentric. By default, the agent sees a square of 7x7 tiles in the direction it is facing. These include the tile the agent is standing on. The agent cannot see through walls or closed doors. The observations are provided as a tensor of shape 7x7x3. However, note that these are not RGB images. Each tile is encoded using 3 integer values: one describing the type of object contained in the cell, one describing its color, and a flag indicating whether doors are open or closed. This compact encoding was chosen for space efficiency and to enable faster training. For all tasks, the agent gets an egocentric view of its surroundings, consisting of 3×3 pixels. A neural network parameterized as a CNN is used to process the visual observation.

The *MultiRoomNXSY* environment consists of X rooms, with size at most Y, connected in random orientations. The agent is placed in the first room and must navigate to a green goal square in the most distant room from the agent. The agent receives an egocentric view of its surrounding, consisting of 3×3 pixels. The task increases in difficulty with X and Y. Episodes finish with a positive reward when the agent reaches the green goal square. Otherwise, episodes are terminated with zero reward after a maximum of 20xN steps.

In the *KeyCorridorS3R3* environment, the agent has to pick up an object which is behind a locked door. The key is hidden in another room, and the agent has to explore the environment to find it. Episodes finish with a positive reward when the agent picks up the ball behind the locked door or after a maximum of 270 steps.

In the *ObstructedMaze2Dlh* environment, the agent has to pick up a box which is placed in a corner of a 3x3 maze. The doors are locked, the keys are hidden in boxes and the doors are obstructed by balls. Episodes finish with a positive reward when the agent picks up the ball behind the locked door or after a maximum of 576 steps.

In the *DynamicObstacles* environment, the agent has to navigate to a fixed goal location while avoiding moving obstacles. In our experiments, the agent is randomly initialized to a location in the grid. If the agent collides with an obstacles, it receives a penalty of -1 and the episode ends.

## A.4    VIZDOOM ENVIRONMENT

We consider the Doom 3D navigation task where the action space of the agent consists of four discrete actions: move forward, move left, move right and no-action. Our testing setup in all the experiments is the *DoomMyWayHome-v0* environments which is available as part of OpenAI Gym (Brockman et al., 2016). Episodes are terminated either when the agent finds the vest or if the agent exceeds a maximum of 2100 time steps. The map consists of 9 rooms connected by corridors and the agent is tasked to reach some fixed goal location from its spawning location. The agent is always spawned in Room-13 which is 270 steps away from the goal under an optimal policy. A long sequence of actions is required to reach the goals from these rooms, making this setting a hard exploration problem. The agent is only provided a sparse terminal reward of +1 if it finds the vest and 0 otherwise. While this environment has sparse reward, it is not procedurally-generated, so the agent finds itself in exactly the same environment in each episode and does not need to generalize to different environment instantiations. This environment is identical to the "sparse" setting used in Pathak et al. (2017).

## A.5    ABLATIONS

In this section, we aim to better understand the effect of using episodic discounting as part of the intrinsic reward, as well as that of using entropy regularization as part of the IMPALA loss.

Figure 9 compares the performance of our model on different MiniGrid tasks with that of three ablations. The first one only uses episodic state counts as exploration bonus without multiplying it by the impact-driven intrinsic reward (*OnlyEpisodicCounts*), the second one only uses the impact-driven exploration bonus without multiplying it by the episodic state count term (*NoEpisodicCounts*), while the third one is the *NoEpisodicCounts* model without the entropy regularization term in the IMPALA loss (*NoEntropyNoEpisodicCounts*).

*OnlyEpisodicCounts* does not solve any of the tasks. *NoEntropyNoEpisodicCounts* either converges to a suboptimal policy or completely fails. In contrast, *NoEpisodicCounts* can solve the easier tasks but it requires more interactions than RIDE and fails to learn on the hardest domain. During training, *NoEpisodicCounts* can get stuck cycling between two states (with a large distance in the embedding states) but due to entropy regularization, it can sometimes escape such local optima (unlike *NoEntropyNoEpisodicCounts*) if it finds extrinsic reward. However, when the reward is too sparse, *NoEpisodicCounts* is insufficient while RIDE still succeeds, indicating the effectiveness of augmenting the impact-driven intrinsic reward with the episodic count term.

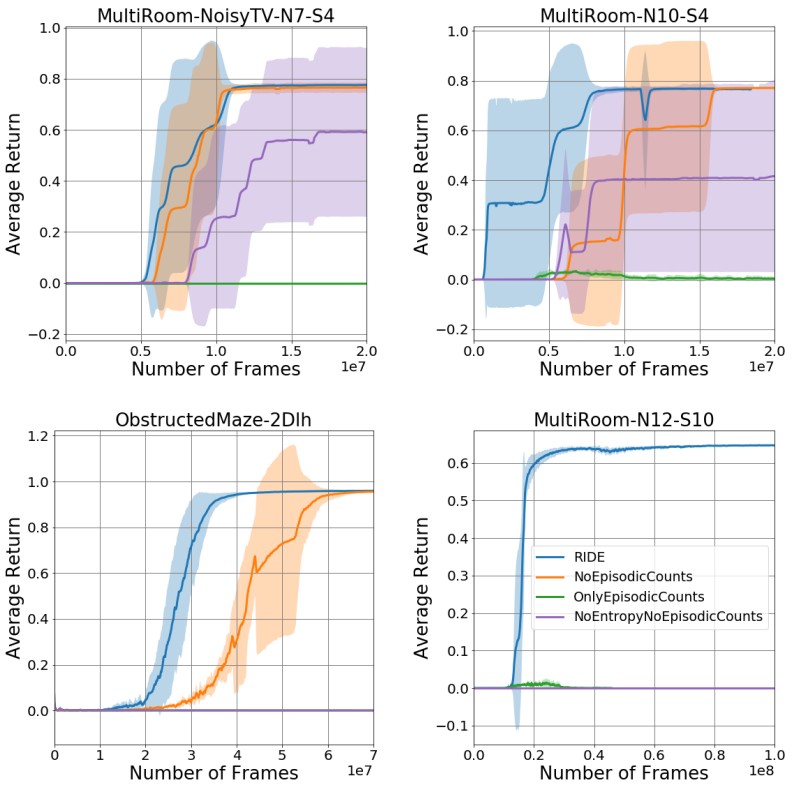

Figure 9: Comparison between the performance of RIDE and three ablations: *OnlyEpisodicCounts*, *NoEpisodicCounts*, and *NoEntropyNoEpisodicCounts*.

Figure 10 shows the average number of states visited during an episode of MultiRoomN12S10, measured at different training stages for our full RIDE model and the *NoEpisodicCounts* ablation. While the *NoEpisodicCounts* ablation always visits a low number of different states each episode ($\leq 10$), RIDE visits an increasing number of states throughout training (converging to $\sim 100$ for an optimal policy). Hence, it can be inferred that *NoEpisodicCounts* revisits some of the states. This claim can be further verified by visualizing the agents' behaviors. After training, *NoEpisodicCounts* goes back and forth between two states, while RIDE visits each state once on its path to the goal. Consistent with our intuition, discounting the intrinsic reward by the episodic state-count term does help to avoid this failure mode.

## A.6 ANALYSIS

### A.6.1 STATE VISITATION IN MULTIROOMN12S10

In this section, we analyze the behavior learned by the agents. Figure 11 shows the state visitation heatmaps for all models trained on 100m frames of MultiRoomN12S10, which has a very sparse reward. Note that while our model has already reached the goal in the farthest room of the maze, Count has explored about half of the maze, while RND and ICM are still in the first two rooms.

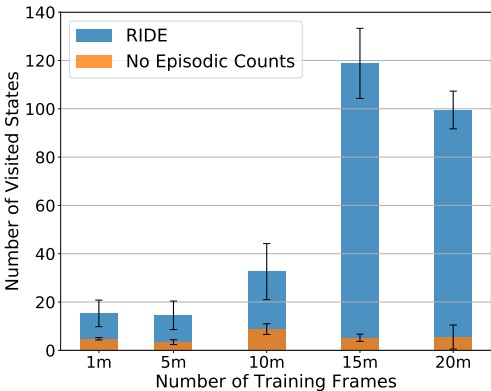

Figure 10: Average number of states visited during an episode of MultiRoomN12S10, measured at different training stages for our full RIDE model (blue) and the *NoEpisodicCounts* ablation (orange).

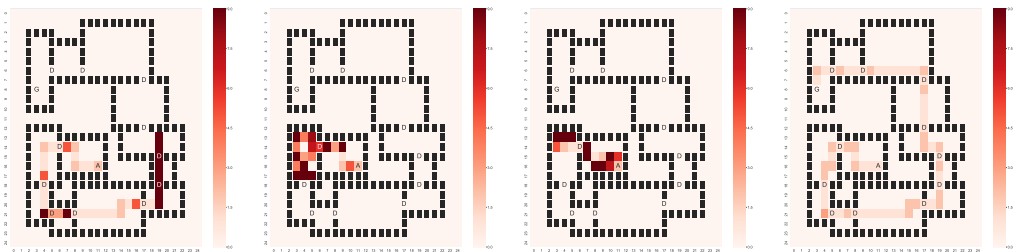

Figure 11: State visitation heatmaps for Count, RND, ICM, and RIDE (from left to right) trained for 100m frames on MultiRoomN12S10.

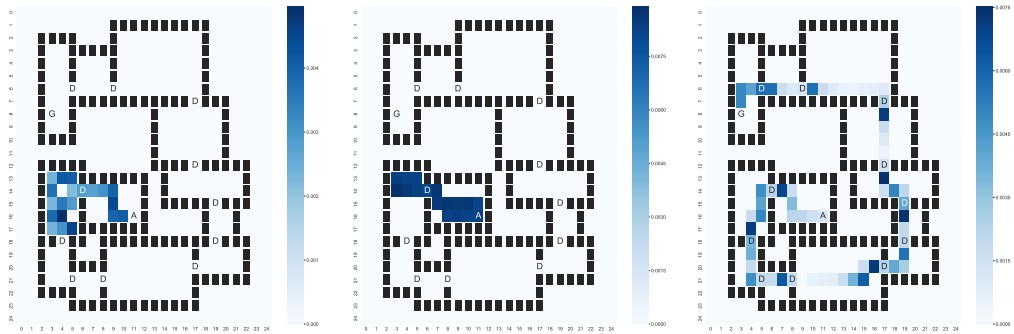

Figure 12: Intrinsic reward heatmaps for RND, ICM, and RIDE (from left to right) on Multi-RoomN12S10.

### A.6.2 INTRINSIC REWARD IN MULTIROOMN12S20

Figure 12 shows a heatmap of the intrinsic reward received by RIDE, RND, and ICM agents trained on the procedurally-generated MultiRoomN12S10 environment. Table 3 shows the corresponding intrinsic rewards received for each type of action, averaged over 100 episodes, for the trained models. This environment is very challenging since the chance of randomly stumbling upon extrinsic reward is extremely low. Thus, we see that while the intrinsic reward provided by RIDE is still effective at exploring the maze and finding extrinsic reward, the exploration bonuses used by RND and ICM are less useful, leading to agents that do not go beyond the second room, even after training on 100m frames.

| Model | Open Door | | Turn Left / Right | | Move Forward | |
|---|---|---|---|---|---|---|
| | Mean | Std | Mean | Std | Mean | Std |
| RIDE | 0.0116 | 0.0011 | 0.0042 | 0.0020 | 0.0032 | 0.0016 |
| RND | 0.0041 | 0.0016 | 0.0035 | 0.0013 | 0.0034 | 0.0012 |
| ICM | 0.0082 | 0.0003 | 0.0074 | 0.0005 | 0.0086 | 0.0002 |

Table 3: Mean intrinsic reward per action computed over 100 episodes on a random map from MultiRoomN12S10.

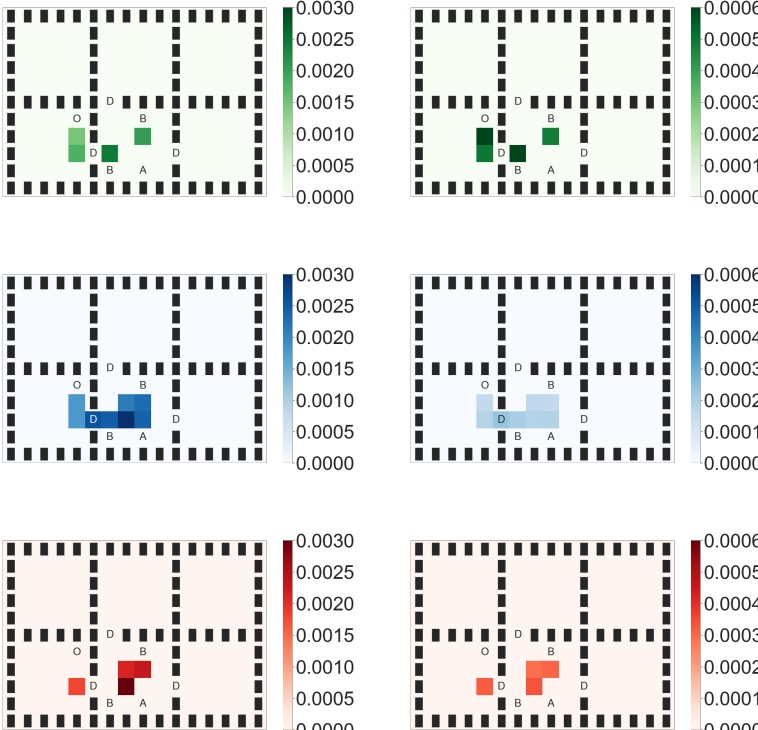

Figure 13: Intrinsic reward heatmaps for RND (left) and RIDE (right) for interacting with objects (i.e. open doors, pick up / drop keys or balls) (green), moving forward (blue), or turning left or right (red) on a random map from ObstructedMaze2Dlh. A is the agent's starting position, K are the keys hidden inside boxes (that need to be opened in order to see their colors), D are colored doors that can only be opened by keys with the same color, and B is the ball that the agent needs to pick up in order to win the game. After passing through the door the agent also needs to drop the key in order to be able to pick up the ball since it can only hold one object at a time.

### A.6.3    INTRINSIC REWARD IN OBSTRUCTEDMAZE2DLH

In order to understand how various interactions with objects are rewarded by the different exploration methods, we also looked at the intrinsic reward in the ObstructedMaze2Dlh environment which contains multiple objects . However, the rooms are connected by locked doors and the keys for unlocking the doors are hidden inside boxes. The agent does not know in which room the ball is located and it needs the color of the key to match that of the door in order to open it. Moreover, the agent cannot hold more than one object so it needs to drop one in order to pick up another.

Figure 13 and Table 4 indicate that RIDE rewards the agent significantly for interacting with various objects (e.g. opening the box, picking up the key, opening the door, dropping the key, picking up the ball) relative to other actions such as moving forward or turning left and right. In contrast, RND again rewards all actions much more uniformly and often times, within an episode, it rewards the interactions with objects less than the ones for moving around inside the maze.

| Model | Open Door | | Pick Ball | | Pick Key | | Drop Key | | Other | |
|---|---|---|---|---|---|---|---|---|---|---|
| | Mean | Std | Mean | Std | Mean | Std | Mean | Std | Mean | Std |
| RIDE | 0.0005 | 0.0002 | 0.0004 | 0.0001 | 0.0004 | 0.00001 | 0.0004 | 0.00007 | 0.0003 | 0.00001 |
| RND | 0.0034 | 0.0015 | 0.0027 | 0.0006 | 0.0026 | 0.0060 | 0.0030 | 0.0010 | 0.0025 | 0.0006 |

Table 4: Mean intrinsic reward per action computed over 100 episodes on a random map from ObstructedMaze2Dlh.

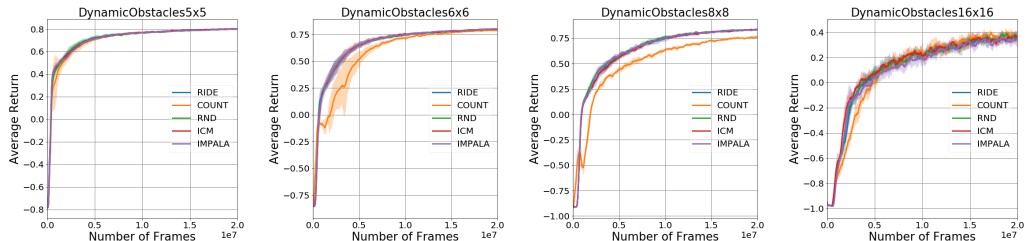

Figure 14: Performance on DynamicObstacles with varying degrees of difficulty.

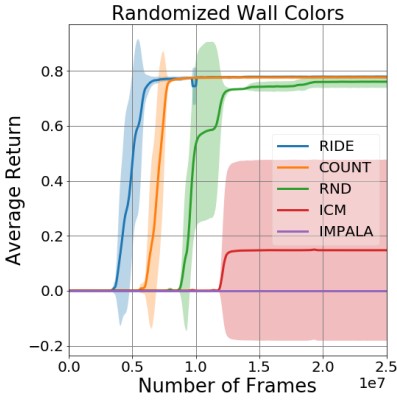

Figure 15: Performance on a version of the MiniGridRoomN7S4 in which the colors of the walls and goals are randomly picked from a set of 4 colors at the beginning of each episode.

## A.7 DYNAMIC OBSTACLES ENVIRONMENT

One potential limitation of RIDE is that it may be drawn to take actions that significantly change the environment, even when those actions are undesirable. In order to test the limits of RIDE, we ran experiments on the DynamicObstacles environment in MiniGrid. As seen in Figure 14, RIDE learns to solve the task of avoiding the moving obstacles in the environment, even if chasing them provides large intrinsic rewards. Hence, RIDE is still able to learn effectively in certain scenarios in which high-impact actions are detrimental to solving the task.

## A.8 GENERALIZATION TO UNSEEN COLORS

In order to test generalization to unseen colors, we also ran experiments on a version of Multi-RoomN7S4 in which the colors of the walls and the goal change at each episode. The models are trained on a set of 4 colors and tested on a held-out set of 2 colors. As seen in Figure 15 and Table 5, RIDE and Count learn to solve this task and can generalize to unseen colors at test time without any extra fine-tuning. RND and ICM perform slightly worse on the test environments, and only one out of five seeds of ICM converges to the optimal policy on the train environments. The best seed for each model was used to evaluate on the test set.

|  | Test Return | |
| --- | --- | --- |
| **Model** | **Mean** | **Std** |
| RIDE | 0.77 | 0.02 |
| Count | 0.77 | 0.02 |
| RND | 0.76 | 0.11 |
| ICM | 0.73 | 0.03 |
| IMPALA | 0.00 | 0.00 |

Table 5: Average return over 100 episodes on a version of MultiRoomN7S4 in which the colors of the walls and goals change with each episode. The models were trained until convergence on a set of 4 colors and tested on a held-out set of 2 colors.

## A.9 OTHER PRACTICAL INSIGHTS

While developing this work, we also experimented with a few other variations of RIDE that did not work. First, we tried to use observations instead of learned state embeddings for computing the RIDE reward, but this was not able to solve any of the tasks. Using a common state representation for both the policy and the embeddings also proved to be ineffective.

