# OpenReview forum: "RIDE: Rewarding Impact-Driven Exploration for Procedurally-Generated Environments"
_ICLR.cc/2020/Conference — Accept (Poster)_

### Official Review · AnonReviewer2 · 2019-10-23
**Official Blind Review #2**

**Rating:** 8

**Review:**

The paper addresses the problem of intrinsically motivating in DRL. In particular, it focuses on exploration of procedurally generated environments where many states are novel compared to training experiences. It offers an intrinsic reward based on large movement in a state embedding space where this state embedding representation is co-trained on the same data already collected for learning. The paper claims to overcome shortcomings of specific past approaches (e.g. count-based / curiosity).

The need for intrinsic motivation in exploration is well motivated, and the approach for training a state embedding is anchored in multiple past works. The use of movement in this state embedding as an intrinsic reward is importantly novel and valuable. The problematic propensity for RL researchers to train on the test environments or design agents that are confused by proverbial noisy TVs and/or sacrifice extrinsic rewards in favor of intrinsic rewards is satisfyingly discussed and addressed through detailed experiments.

This reviewer moves to accept the paper for its contributions to intrinsically motivated exploration with thorough discussion of how the technique addresses shortcomings of past methods. This reviewer is thankful that the authors do not overinterpret the MiniGrid results and that they provide intuition for why the state embedding functions capture what we want them to capture. The fact that this approach makes joint use of the whole (s,a,r,s') tuple feels significant, as does the fact that this approach does not require any changes to the policy network (e.g. presuming that features useful for computing intrinsic rewards are also going to be useful for directly acting to optimize extrinsic rewards).

Question:
- In partially observable environments that require agents to wait for something, should a RIDE-motivated agent consider changes in its own internal clocks (part of the recurrent state) impactful moves? If an environment might require a recurrent / history-aware action policy, should RIDE also be made history aware? Might a history-aware RIDE reward sufficiently motivate a stateless/reactive policy?

**Experience Assessment:**

I have published one or two papers in this area.

**Review Assessment: Checking Correctness Of Derivations And Theory:**

N/A

**Review Assessment: Checking Correctness Of Experiments:**

I assessed the sensibility of the experiments.

**Review Assessment: Thoroughness In Paper Reading:**

I read the paper at least twice and used my best judgement in assessing the paper.

---

> ### Author Response · Authors · 2019-11-09
> **Response to Review #2**
>
> We thank the reviewer for the positive feedback and are happy to hear they found our work “importantly novel and valuable”, containing “detailed experiments” and a “thorough discussion of how the technique addresses shortcomings of past methods”.
>
>
> “In partially observable environments that require agents to wait for something, should a RIDE-motivated agent consider changes in its own internal clocks (part of the recurrent state) impactful moves?...”
>
> These questions open up exciting avenues for future work. We have recently began to explore similar ideas in which the embeddings are learned using recurrent networks (instead of feed-forward ones), but we do not have conclusive answers to the above questions yet. We believe these would be better addressed as a separate contribution.

---

### Official Review · AnonReviewer3 · 2019-10-24
**Official Blind Review #3**

**Rating:** 6

**Review:**

This paper proposes a new intrinsic reward method for model-free reinforcement learning agents in environments with sparse reward. The method, Impact-Driven Exploration, learns a state representation of the environment separate from the agent to be trained, based on a combied forward and inverse dynamics loss. The agent is then separately trained with a reward encouraging sequences of actions that maximally change the learned state.

Like other latent state transition models (Pathak et al. 2017), RIDE learns a state representation based on a combined forward and inverse dynamics loss. However, Pathak et al. rewards the agent for taking actions that lead to large difference between the actual next state and the predicted next state. RIDE instead rewards the agent for taking actions that lead to a large difference between the actual next state and the current state. However, because rewarding one-step state differences may cause an agent to loop between two maximally-different states, the RIDE loss term is augmented with a state visitation count term, which decreases intrinsic reward for a state based on the number of times that state has been visited in the current episode.

The experiments compare RIDE to a selection of other intrinsic reward methods in the MiniGrid, Mario, and VizDoom environments. RIDE provides improved performance on a number of tasks, and solves challenging versions of the MiniGrid tasks that are not solved by other algorithms.

Decision: Weak Accept.

The main weakness of the paper seems to be a limitation in novelty.
Previous papers such as (Pathak et al. 2017) have trained RL policies using an implicit reward based on learned latent states. Previous papers such as (Marino et al. 2019) have used difference between subsequent states as an implicit reward for training an RL policy. It is not a large leap to combine these two ideas by training with difference between subsequent learned states. However, this paper seems to be the first to do so.

Strengths:
The experiments section is very thorough, and the visualizations of state counts and intrinsic reward returns are insightful.
The results appear to be state of the art for RL agents on the larger MiniGridWorld tasks.
The paper is clearly-written and easy to follow.
The Mario environment result discussed in section 6.2 is interesting in its own right, and provides some insight into previous work.

Despite the limited novelty of the IDE reward term, the experiments and analysis provide insight into the behavior of trained agents and the results seem to improve on existing methods.
Overall, the paper seems like a worthwhile contribution.

Notes:
In section 2 paragraph 4, "sintrinsic" should be "intrinsic".
In section 3, at "minimizes its discounted expected return," seems like it should be "maximizes".
The explanation of IMPALA (Espeholt et al., 2018) should occur before the references to IMPALA on page 5.
Labels for the axes in figures 4 and 6 would be helpful for readability.

The motivation for augmenting the RIDE reward with an episodic count term is that the IDE loss alone would cause an agent to loop between two maximally different states.
It would be interesting to know whether this suspected behavior actually occurs in practice, and how much the episodic count term changes this behavior.
It is surprising that in the ablation in section A.5, removing the state count term does not lead to the expected behavior of looping between two states, but instead the agent converges to the same behavior as without the state count term.

Also, in Figure 9, was the OnlyEpisodicCounts ablation model subjected to the same grid search described in A.2, or was it trained with the same intrinsic reward coefficient as the other models?
Based on the values in Table 4, it seems like replacing the L2 term with 1 without changing the reward coefficient would multiply the intrinsic reward by a large value.


**Experience Assessment:**

I have read many papers in this area.

**Review Assessment: Checking Correctness Of Derivations And Theory:**

I assessed the sensibility of the derivations and theory.

**Review Assessment: Checking Correctness Of Experiments:**

I carefully checked the experiments.

**Review Assessment: Thoroughness In Paper Reading:**

I read the paper thoroughly.

---

> ### Author Response · Authors · 2019-11-09
> **Response to Review #3**
>
> We thank the reviewer for the detailed and thoughtful comments. We appreciate they consider our work to be a “worthwhile contribution”, our experimental section “very thorough” and our visualizations “insightful”.
>
>
> “The motivation for augmenting the RIDE reward with an episodic count term is that the IDE loss alone would cause an agent to loop between two maximally different states.
> It would be interesting to know whether this suspected behavior actually occurs in practice, and how much the episodic count term changes this behavior.”
>
> We thank the reviewer for suggesting to investigate this question in more detail. We carried out additional analyses and updated the draft. We have found that this behavior does occur in practice and can be observed by visualizing the agents’ trajectories. After training on the MultiRoom-N12-S10 task, the NoEpisodicCounts ablation visits two of the states a large number of times going back and forth between them, while RIDE visits each state once on its path to the goal.
>
> Figure 10 in the Appendix further supports this claim by showing the number of different states the agent visits within an episode. While the NoEpisodicCounts ablation always visits a low number of different states (~< 10) each episode, RIDE visits an increasing number of states throughout training (converging to ~100 for an optimal policy). From this, we can infer that  NoEpisodicCounts revisits some of the states.
>
>
> “It is surprising that in the ablation in section A.5, removing the state count term does not lead to the expected behavior of looping between two states, but instead the agent converges to the same behavior as without the state count term.”
>
> The agent is also encouraged to explore different (state, action) pairs via the entropy regularization term in the IMPALA loss, which can help avoid local optimum in certain cases. During training, the NoEpisodicCounts agent exhibits the behavior of looping between two states, but due to entropy regularization, it can get unstuck once it finds some extrinsic reward. This can explain why the NoEpisodicCounts ablation takes longer to converge than RIDE, which is less prone to this cycling behavior. Note that in the more challenging MultiRoom-N12-S10 environment, NoEpisodicCounts does not learn a useful policy, likely because the extrinsic reward is too sparse, so the agent remains stuck in a cycle.
>
> To further support the above hypothesis, we have added experiments with the NoEpisodicCounts model without entropy regularization. The results show that without the entropy loss term, NoEpisodicCounts is more likely to completely fail or converge to a suboptimal policy.
>
>
> “Also, in Figure 9, was the OnlyEpisodicCounts ablation model subjected to the same grid search described in A.2, or was it trained with the same intrinsic reward coefficient as the other models?”
>
> We ran the same grid search for the OnlyEpisodicCounts ablation.
>
>
> “Based on the values in Table 4, it seems like replacing the L2 term with 1 without changing the reward coefficient would multiply the intrinsic reward by a large value.”
>
> We are unsure of what you mean by “replacing the L2 term with 1”. Could you kindly clarify the question?

---

### Official Review · AnonReviewer1 · 2019-10-26
**Official Blind Review #1**

**Rating:** 6

**Review:**

Summary
This paper proposes a Rewarding Impact-Driven Exploration (RIDE), which is an intrinsic exploration bonus for procedurally-generated environments. RIDE is built upon the ICM architecture (Pathak et al. 2017), which learns a state feature representation by minimizing the L2 distance between the actual next state feature and the predicted next state feature while minimizing the cross-entropy loss between the true action and the estimated action from the consecutive state features. Finally, RIDE's intrinsic reward bonus is computed by L2 norm of the difference between the current state feature and the next state feature, divided by the square root of the visitation count of the next state within the episode. Experimental results show that RIDE outperforms the existing exploration methods in the procedurally-generated environments (MiniGrd), and is competitive in singleton environments.


Comments and questions:
- In reinforcement learning, the agent should explore the experiment due to uncertainty. If everything in the environment is certain to the agent, then it does not have to explore and just exploiting the past experience would be the best. My major concern about the paper is 'impact-driven' reward bonus may not account for the uncertainty. Constantly encouraging the states that have a high impact would not always good, and it may interfere to converge to an optimal policy.
- It seems that RIDE assumes that 'high-impact' states are always good, thus rewarded. It could be true on the conducted MiniGrid domains, but this assumption may not hold in general. Could 'impact-driven' exploration be realistic and be applied to more general problems?
- Similarly, in the problems where high-impact states have to be avoided, can RIDE still work effectively? For example, how about 'Dynamic-Obstacles' domains implemented in MiniGrid? In this task, RIDE may promote to chase obstacles that have to be avoided, interfering with learning optimal policy. It would be great to show the effectiveness of RIDE in such environments.
- In MiniGrid problems, if the colors of walls and the goal are changed at every episode, does RIDE work well?
- In Figure 4, why the intrinsic reward heatmaps are drawn only on the straight paths?
- Minor: In the last sentence of Section 3, "the current state and the next state predicted by the forward model" -> "the actual next state and the next state predicted by the forward model"


---
after rebuttal:

Thank the authors for clarifying my questions and concerns. Most of my concerns are addressed, and I raise my score accordingly.

**Experience Assessment:**

I have read many papers in this area.

**Review Assessment: Checking Correctness Of Derivations And Theory:**

N/A

**Review Assessment: Checking Correctness Of Experiments:**

I carefully checked the experiments.

**Review Assessment: Thoroughness In Paper Reading:**

I read the paper at least twice and used my best judgement in assessing the paper.

---

> ### Author Response · Authors · 2019-11-09
> **Response to Review #1**
>
> We thank the reviewer for their time and feedback.
>
> “In reinforcement learning, the agent should explore the experiment due to uncertainty. If everything in the environment is certain to the agent, then it does not have to explore and just exploiting the past experience would be the best. My major concern about the paper is 'impact-driven' reward bonus may not account for the uncertainty. ”
>
> While we agree that uncertainty estimation has been useful for developing exploration methods in the past, one of our main findings is that such methods can in fact be ineffective in certain settings. It seems that existing methods estimate the uncertainty poorly in sparse-reward partially-observed procedurally-generated environments. In such environments, the dynamics can be learned early in training without being helpful in guiding exploration towards extrinsic rewards in the environment. For example, in Fig 7 we demonstrate that the intrinsic reward of the ICM, RND, and Count methods diminishes very fast during training, suggesting that the agent has a good model of the transition dynamics (ICM) or has seen similar states before (RND, Count) so its uncertainty about the world is low, yet it fails to solve the task because it  hasn’t found extrinsic reward. We believe the MiniGrid environments used in our work present a more challenging and realistic setting than previously used environments that are fully observable or do not change across episodes.
>
>
> “Constantly encouraging the states that have a high impact would not always good, and it may interfere to converge to an optimal policy...”
> “It seems that RIDE assumes that 'high-impact' states are always good, thus rewarded...”
>
> Thank you for your question. While we agree that there exist settings in which certain “high-impact” actions may not help the agent to solve a task, we believe there are a few ways in which this issue is already addressed in our current formulation. First, the agent also learns from extrinsic reward, so if that action is negatively correlated with the extrinsic reward, the agent can learn, in principle, to avoid that action. Second, the agent also explores via entropy regularization, which can help to avoid getting stuck in a local optimum. For example, the MultiRoom-NoisyTV environment contains a high-impact action that is not useful for solving the task and it isn’t penalized by negative extrinsic reward. Even in this more challenging setting, RIDE learns an optimal policy.
>
>
> “Similarly, in the problems where high-impact states have to be avoided, can RIDE still work effectively? For example, how about 'Dynamic-Obstacles' domains implemented in MiniGrid?...”
>
> We have updated the paper with experiments on Dynamic-Obstacles. RIDE learns to avoid the obstacles and reach the goal.
>
>
> “In MiniGrid problems, if the colors of walls and the goal are changed at every episode, does RIDE work well?”
>
> We added experiments for answering this question in the revised draft. RIDE learns to solve this task and can even generalize to unseen colors at test time without any further fine-tuning.
>
>
> “In Figure 4, why the intrinsic reward heatmaps are drawn only on the straight paths?”
>
> Figure 4 shows the trajectories of fully-trained models on MultiRoom-N7-S4. On this task, all agents learn optimal policies, so their behavior follows a shortest path to the goal.

---

### Author Response · Authors · 2019-11-09
**Paper Update**

We thank all the reviewers for their constructive feedback.

We have updated the paper with the following:

    1. Experiments on 4 settings with varying degrees of difficulty in the Dynamic-Obstacles environment (see Appendix A.7 and Figure 14).
    2. Experiments on a modified version of MultiRoom-N7-S4 in which the colors of the walls and the goal change at every episode. We also evaluate the models on a held-out set of colors (see Appendix A.8, Figure 15, and Table 5).
    3. Extra qualitative and quantitative analysis on the effects of augmenting the intrinsic reward with the episodic count term, comparing RIDE with the NoEpisodicCounts ablation (see Appendix A.5 and Figure 10).
    4. Experiments with NoEpisodicCounts without entropy regularization to better understand the effect of the entropy loss term on avoiding local optima (see Appendix A.5 and Figure 9).

We also made minor corrections to the text taking into account reviewers’ suggestions.

---

### Decision · Program_Chairs · 2019-12-19

**Decision:**

Accept (Poster)

**Comment:**

This paper tackles the problem of exploration in deep reinforcement learning in procedurally-generated environments, where the same state is rarely encountered twice. The authors show that existing methods do not perform well in these settings and propose an approach based on intrinsic reward bonus to address this problem. More specifically, they combine two existing ideas for training RL policies: 1) using implicit reward based on latent state representations (Pathak et al. 2017) and 2) using implicit rewards based on difference between subsequent states (Marino et al. 2019).

Most concerns of the reviewers have been addressed in the rebuttals. Given that it builds so closely on existing ideas, the main weakness of this work seems to be the novelty. The strength of this paper resides in the extensive experiments and analysis that highlight the shortcomings of current techniques and provide insight into the behaviour of trained agents, in addition to proposing a strategy which improves upon existing methods.

The reviewers all agree that the paper should be accepted. I therefore recommend acceptance.